# Regional Impact of COVID-19-Associated Pulmonary Aspergillosis (CAPA) during the First Wave

**DOI:** 10.3390/jof8020096

**Published:** 2022-01-19

**Authors:** Robbert G. Bentvelsen, Andreas L. E. Van Arkel, Tom A. Rijpstra, Merijn K. M. Kant, Simone Van Der Sar-Van Der Brugge, Daan W. Loth, Peter Van Wijngaarden, Arthur W. F. Du Mée, David C. Y. Yick, Bram M. W. Diederen, Peter C. Wever, Alexander C. A. P. Leenders, Laura Van Dommelen, Klaas H. De Groot, Wouter Van den Bijllaardt, Paul E. Verweij

**Affiliations:** 1Microvida Laboratory for Microbiology, Amphia Hospital, 4818 CK Breda, The Netherlands; avanarkel@amphia.nl (A.L.E.V.A.); wbijllaardt@amphia.nl (W.V.d.B.); 2Department of Medical Microbiology, Leiden University Medical Centre, 2300 RC Leiden, The Netherlands; 3Microvida Laboratory for Microbiology, ETZ, 5000 LE Tilburg, The Netherlands; 4Department of Intensive Care Medicine, Amphia Hospital, 4818 CK Breda, The Netherlands; trijpstra@amphia.nl (T.A.R.); mkant@amphia.nl (M.K.M.K.); 5Department of Pulmonary Medicine, Amphia Hospital, 4818 CK Breda, The Netherlands; svandersar@amphia.nl (S.V.D.S.-V.D.B.); DLoth@amphia.nl (D.W.L.); 6Department of Internal Medicine, Amphia Hospital, 4818 CK Breda, The Netherlands; pvanwijngaarden@amphia.nl; 7Department of Radiology, Amphia Hospital, 4818 CK Breda, The Netherlands; adumee@amphia.nl; 8Department of Pathology, Amphia Hospital, 4818 CK Breda, The Netherlands; dyick@amphia.nl; 9Microvida Laboratory for Microbiology, Bravis Hospital, 4704 RA Roosendaal, The Netherlands; b.diederen@bravis.nl; 10Department of Medical Microbiology, Bernhoven Hospital, 5406 PT Uden, The Netherlands; P.Wever@jbz.nl; 11Department of Medical Microbiology and Infection Control, Jeroen Bosch Hospital, 5223 GZ Hertogenbosch, The Netherlands; a.leenders@jbz.nl; 12Department of Medical Microbiology, Stichting PAMM, 5500 AA Veldhoven, The Netherlands; l.van.dommelen@pamm.nl; 13Department of Intensive Care, Máxima Medical Center, 5631 BM Veldhoven, The Netherlands; klaas.de.groot@mmc.nl; 14Department of Medical Microbiology, Radboud University Medical Center, 6500 HB Nijmegen, The Netherlands; paul.verweij@radboudumc.nl; 15Center of Expertise in Mycology Radboudumc/CWZ, 6252 AG Nijmegen, The Netherlands

**Keywords:** COVID-19, pulmonary aspergillosis, CAPA, corticosteroids, voriconazole

## Abstract

Background: Critically ill COVID-19 patients have proven to be at risk for developing invasive fungal infections. However, the incidence and impact of possible/probable COVID-19-associated pulmonary aspergillosis (CAPA) in severe COVID-19 patients varies between cohorts. We aimed to assess the incidence, risk factors, and clinical outcome of invasive pulmonary aspergillosis in a regional cohort of COVID-19 intensive care patients. Methods: We performed a regional, multicentre, retrospective cohort study in the intensive care units (ICUs) in North Brabant, The Netherlands. We included adult patients with rt-PCR-confirmed SARS-CoV-2 infection (COVID-19), requiring mechanical ventilation for acute respiratory distress syndrome. Demographics, clinical course, biomarker value, and treatment outcomes were compared between the groups with possible/probable CAPA from the main study centre and the regional centres, and without signs of CAPA from the main study centre as controls. The primary aim was to assess the regional impact of possible/probable CAPA in COVID-19 ICU patients, measured as all-cause mortality at 30 days after ICU admission. Secondary outcomes were risk factors for developing CAPA, based on underlying host factors and to identify the value of the mycological arguments for the diagnosing of CAPA. Results: Between 1 March and 30 April 2020, we included 123 patients with severe COVID-19: 29 patients (30.9%) in the main ICU with possible/probable CAPA, and 65 (69.1%) with no signs of CAPA; 29 patients in the regional ICUs with signs of CAPA. Patients’ characteristics and risk factors did not differ for CAPA and non-CAPA patients. Patients with COPD and/or chronic steroid medication developed CAPA more frequently, although this was not statistically significant. CAPA patients were admitted to the ICU earlier, had lower PF-ratios, and more often required renal replacement therapy. All-cause 30-day mortality was significantly higher in mechanically ventilated COVID-19 patients with possible/probable CAPA 39.7% (23/58) compared to patients without evidence for CAPA 16.9% (11/65) (OR 3.2 [95% CI 1.4–7.4] *p* = 0.005). Conclusion: The high incidence of possible and probable CAPA in critically ill COVID-19 patients is alarming. The increase in 30-day mortality in CAPA highlights the need for active surveillance and management strategies in critically ill COVID-19 patients.

## 1. Introduction

The pandemic of Coronavirus Disease 2019 (COVID-19), caused by the severe acute respiratory syndrome coronavirus 2 (SARS-CoV-2), has high attack rates, intensive care unit (ICU) admissions, and mortality. Substantial case fatality rate is reported in patients admitted to the ICU, of which half developed secondary infections [1]. Late February 2020, the province of North-Brabant in the Netherlands emerged as a hotspot for COVID-19 [2]. In April 2020, we reported the first remarkable number of cases of invasive pulmonary aspergillosis (IPA) occurring in COVID-19 patients admitted to the ICU, with the first azole-resistant *Aspergillus fumigatus* isolate [3]. Early international warnings and case series also reported on COVID-19-associated pulmonary aspergillosis (CAPA) [4,5,6,7]. Dramatic high rates were noted in COVID-19 ICU patients of possible/probable CAPA (19–33%) with high mortality rates (44–66%) [3,5,6].

During the first wave of COVID-19 in Europe, the clinical consequences of screening for mycological arguments in critically ill patients with COVID-19 were not clear. The ECMM/ISHAM Consensus CAPA case definition of mid 2020 enables classification of cases and comparison of studies [8]. The incidence of CAPA in critically ill patients varies in major cohorts from the United Kingdom (14.1%), Italy (27.7%), and France (19.6%) [9,10,11].

After the first description of a case series of six patients with possible/probable CAPA, in this study we extended the cohort to include possible/probable CAPA cases in COVID-19 patients from ICU’s of all the regional medical centres in North Brabant, The Netherlands during the first wave of COVID-19. We aimed to assess the regional impact of CAPA in COVID-19 ICU patients, primarily measured as all-cause mortality at 30 days after ICU admission. Our secondary objectives were to define risk factors for developing CAPA based on underlying host factors, such as comorbidity, and to identify the value of the mycological arguments for diagnosing CAPA.

## 2. Methods

### 2.1. Study Design and Population

We conducted a regional multi-centre, retrospective cohort study over the months March and April 2020. All adult patients with laboratory confirmed COVID-19 admitted to the ICU of the main study centre (Amphia hospital, Breda), with and without mycological arguments for CAPA were assessed. At the other regional study centres in the province of North Brabant, adult ICU patients with mycological arguments for CAPA were included. Groups were compared as A: Amphia CAPA, B: Brabant CAPA, and C: Amphia Controls (non-CAPA).

Patients were eligible for inclusion if they were 18 years or older with rt-PCR confirmed SARS-CoV-2 infection (COVID-19) and admitted to the ICU of one of the study centres, with the first day of ICU admission within the study period. Cases were excluded if the patient opted for their medical data not to be used, or was outside of the main study centre without any mycological arguments for CAPA. In the main study centre, COVID-19 patients routinely underwent bronchoscopy in the second week of ICU admission, prior to the start of corticosteroid therapy at deterioration without known cause.

Primary endpoint was the all-cause mortality at 30 days after ICU admission with CAPA compared to the non-CAPA population of COVID-19 patients with ICU admission. We also assessed electronic health records for co-morbidities, inflammatory biomarkers, mycological arguments, bronchoscopy findings, radiological imaging, and effect of antifungal treatment of possible/probable CAPA. For both the primary and secondary endpoints the time (in days after ICU admission) at which the endpoint was reached was noted. Patients are classified according to the ECMM/ISHAM consensus criteria [8].

### 2.2. Recruitment and Consent

Due to the observational nature of this study, participants were not subjected to any procedures, nor will the care they receive be influenced by enrolment in this trial. Therefore, the Dutch Medical Research Involving Human Subjects Act (WMO) is not applicable to this study. Because of this, written informed consent from participants is not required before enrolment. However, an opt-out system was employed, where patients’ medical data was not used in this study when the participant, their relatives, and/or their legal representatives object to the use of their (encoded) medical data. This study was conducted according to the principles of the Declaration of Helsinki (version 10, 19 October 2013) and the principles of Good Clinical Practice (GCP).

### 2.3. Statistical Analysis

Descriptive statistics were calculated for all study parameters. Continuous data are described as median with interquartile range (IQR). Categorical variables were described as numbers and percentages. The Fisher’s exact test and the Mann–Whitney U test was used to compare independent continuous or categorical variables when appropriate. Survival curves were constructed with the Kaplan–Meier method and were compared via the log rank test. Survival at 30 days after ICU admission was compared between the possible/probable CAPA and non-CAPA groups. Missing data was coded as missing. In all instances, a *p*-value of less than 0.05 was considered statistically significant. We calculated the difference in all-cause mortality between the CAPA and non-CAPA group in percentages and compared both groups using the Fisher’s exact test.

## 3. Results

During the first wave of COVID-19, 1 March to 30 April 2020, in the main study centre a total of 94 ICU patients with COVID-19 were included (Table 1 and Table 2). Twenty-nine of these patients had signs of CAPA (Group A), of which 23.4% (22/94) were classified as probable CAPA and 7.4% (7/94) as possible CAPA. The control group (C) consisted of 65 (69.1%) ICU patients without evidence for CAPA. A total of 29 patients with possible/probable CAPA were included from the other regional North Brabant centres (B).

The median age of patients with possible/probable CAPA was 69 years (IQR 60–74) and 67 years (IQR 58–74) in the control group. The proportion of males in the possible/probable CAPA group was 69.0% (40/58), in the control group 73.8% (48/65) was male.

Age, sex, BMI, medical history, and past or current smoking did not significantly differ for CAPA and non-CAPA patients. Patients with CAPA more often had COPD and were on systemic corticosteroid medication for at least 30 days prior to ICU admission. These differences were statistically significant in the main study centre and showed the same trend in the overall CAPA versus control group.

Following symptom onset, CAPA patients were admitted to hospital after a median of 7 days (IQR 4–10), and to ICU after 9 (IQR 7–13) days. The severe COVID-19 patients without CAPA were admitted to ICU after a median of 11 days (IQR 9–15), this was significantly longer than CAPA patients. Also, CAPA patients had comparable SOFA scores on ICU day 1 and max. The ARDS in CAPA patients was worse with the median lowest Horowitz score 123 mmHg (IQR 89–156) versus 157 mmHg (IQR 120–200) in non-CAPA. During ICU admission CAPA patients required renal replacement therapy more frequently (46.9% v. 15.9%, *p* = 0.001).

*Aspergillus fumigatus* was cultured in 44.8% (26/58) of possible/probable CAPA patients. These cultures were performed on sputum samples, tracheal aspirates, and broncho-alveolar lavage (BAL) fluids. Galactomannan (GM) enzyme-linked immunosorbent assay (ELISA) was positive in 55.2% of patients (32/58). GM ELISA was performed on tracheal aspirates, BAL fluid, serum, and sputum (two patient samples). GM lateral flow assay (LFA) was positive in 48.3% of patients (28/58). GM LFA was performed on sputum samples, tracheal aspirates, and BAL fluids. In group A, fungal culture was positive in 22.2% (6/27) of the BAL/bronchial aspirate/sputum samples with positive GM assays (GM ELISA and LFA).

Of the possible/probable CAPA cases, 72.4% (42/58) received anti-fungal treatment, of which 88.1% (37/42) received a combination of voriconazole and anidulafungin; 2.4% (1/42) received voriconazole and caspofungin, in line with the national Dutch guidelines; 2.4% (1/42) of patients were treated with liposomal amphotericin-B; and 7.1% (3/42) were treated with anidulafungin, voriconazole, and liposomal amphotericin-B during the course of their treatment.

Mortality rate at 30 days after ICU admission was 16.9% (11/65) in mechanically ventilated COVID-19 patients without CAPA, and 39.7% (23/58) in the CAPA patients. CAPA was significantly associated with all-cause 30-day mortality (odds ratio [OR] 3.2 [95%CI 1.4–7.4] *p* = 0.005). Survival analysis with the Kaplan–Meier method, confirmed the difference in 30-day mortality after ICU admission, between CAPA and non-CAPA groups (Figure 1) and these groups in the main study centre (Figure 2).

The patients with probable CAPA had an increased chance to die within 30 days compared to patients with possible CAPA or without mycological signs of CAPA. Mortality was greater even in patients with *Aspergillus fumigatus* cultured (Table 3).

## 4. Discussion

This study is an extension of our previous study [2], with a more extensive analysis of patients with possible/probable CAPA compared to severe COVID-19 controls without mycological arguments for CAPA. We found a significant difference in mortality between the possible/probable CAPA and control group. Invasive fungal infection in the possible/probable CAPA group is a probable cause for this difference. Patients with COPD and/or chronic steroid medication more often developed presumptive CAPA. This is in line with previous reports on risk factors of chronic respiratory disease and the use of corticosteroids [10]. However, in this cohort risk factors like BMI, age, and sex did not differ between groups. Mycological diagnosis was often made on BAL fluid GM, where 44% of patients had a positive fungal culture. We observed a 30-day mortality rate of 39.7% in patients with possible/probable CAPA. This percentage is comparable to mortality rates as reported in other studies (38–44%) [9,10].

Strengths of our cohort are the size of the groups, the classification of risk factors and the predefined inclusion period. This provides the best insights in the actual estimates for the impact, minimizing the risks of observer bias. Our experiences with the first few cases did have an effect on the diagnosing and management strategy, so this is a limitation [3,12]. The inclusion of patients with signs of CAPA from all regional ICUs is a strength and gives more insight into the total impact in the province.

One limitation of the comparison between severe COVID-19 patients, with and without signs of CAPA, was that in the CAPA group the patients had a worse clinical course. The higher frequency of renal replacement therapy and pulmonary damage indicated by lowest PaO_2_/FiO_2_ (PF) ratio, could be a confounder or attributable to the secondary fungal infection. On the other hand, SOFA scores on ICU day one did not differ nor did the SOFA max on ICU, implying comparable clinical status at ICU admission.

The diagnosis of CAPA remains challenging [13]. The gold standard for CAPA diagnosis is pathological evidence of invasive fungal growth, which is rarely obtained in this patient group and also highly dependent on the site of tissue collection. Determining the difference between *Aspergillus* spp. colonisation and infection with bronchial fungal cultures, GM assays, microscopy, or PCR is challenging. Secondarily, due to Acute Respiratory Distress Syndrome (ARDS) in severe COVID-19, radiological evidence that supports or excludes invasive pulmonary mould disease is commonly not obtained. Because the difference between invasive fungal growth and colonisation is hard to differentiate, recent proposed clinical guidelines for CAPA suggest a low threshold to start antifungal therapy in mechanically ventilated COVID-19 patients who show clinical deterioration with no other obvious explanation [6].

Furthermore, the comparison of mortality rate between possible/probable CAPA in the main study centre and the remaining study centres was remarkably higher in the main study centre: 51.7% versus 27.6%. In the main study centre, bronchoscopy-obtained samples were available for almost all patients to perform fungal culture and GM assay. Diagnostic tests could have been performed more frequently in the main study centre, on patients with a worsening clinical condition, which may have led to selection bias of patients with a bad prognosis. However, the reported CAPA mortality corresponds to the largest ICU-CAPA study in the literature to date (62%) [11]. Nevertheless, diagnosis and clinical management protocols of CAPA differ between hospitals and could therefore affect outcomes.

Most diagnostic tests for CAPA were performed on appropriate patient samples, however GM ELISA is validated only for serum and BAL fluid. In this study, GM ELISA was performed erroneously on two samples registered as sputum samples. Technically, these samples contain *Aspergillus* spp., however this is not proof for clinically relevant infection. Patients can be colonised with *Aspergillus* spp. in the upper airways, which can lead to false positive clinical results in GM assays [7].

Previous reports demonstrate histology and autopsy confirmed cases of invasive pulmonary aspergillosis in COVID-19 patients [7,14]. Due to the high mortality in possible/probable CAPA patients we suggest a low threshold for the start of antifungal therapy. Treatment of COVID-19 requires a multidisciplinary team (MDT) approach, including the treatment of (suspected) CAPA.

In summary, a significant difference in mortality was found between patients with possible/probable CAPA and controls. Patients with COPD and/or chronic steroid medication developed CAPA more frequently, although this was not statistically significant. CAPA patients were admitted to the ICU earlier, had lower PF-ratios, and more often required renal replacement therapy. However, additional research should be performed into pathogenesis of *Aspergillus species* in patients with COVID-19 pulmonary infection and risk factors should be analysed further. Post mortem studies would give the best understanding of the pathophysiology and impact of CAPA.

## Figures and Tables

**Figure 1 jof-08-00096-f001:**
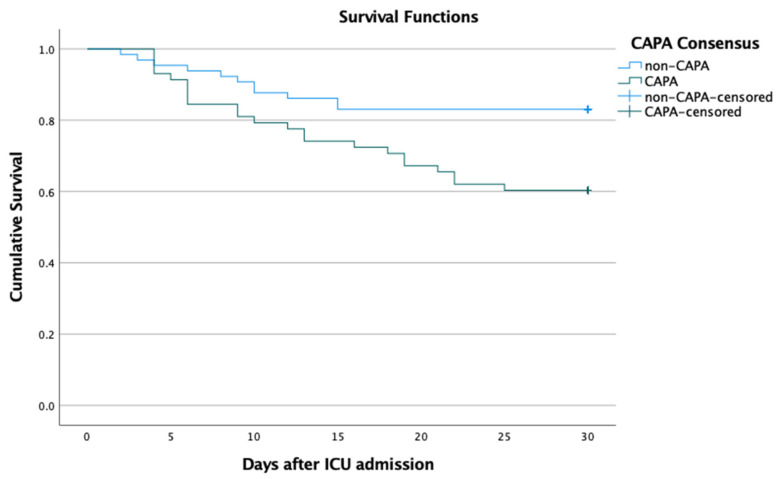
Cumulative 30-day survival of patients with severe COVID-19 fulfilling criteria of probable or possible COVID-19-associated Pulmonary Aspergillosis (CAPA) and patients without CAPA (Control). Log rank *p* = 0.007.

**Figure 2 jof-08-00096-f002:**
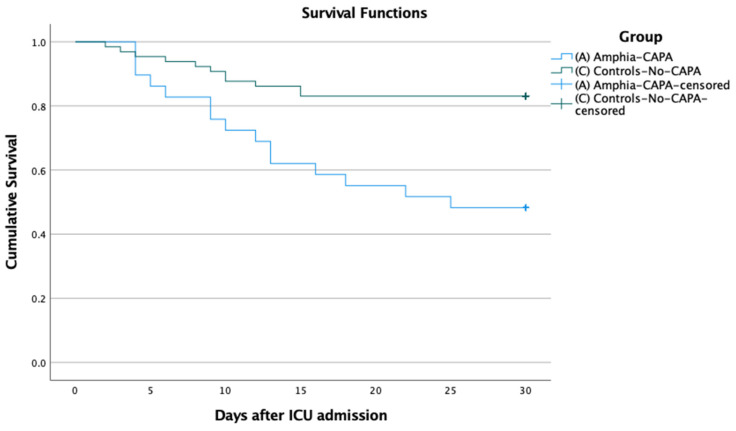
Cumulative 30-day survival of patients with severe COVID-19 fulfilling criteria of probable or possible COVID-19-associated Pulmonary Aspergillosis (CAPA) and patients without CAPA (Control) in the main study centre. Log rank *p* < 0.001.

**Table 1 jof-08-00096-t001:** Patient characteristics, clinical course, and outcome, by aspergillosis status.

	Possible/Probable CAPA		Non-CAPA		*p* Value
	(*n* = 58)		(*n* = 65)		
Male	40	69.0%	48	73.8%	0.549
Age (years)	69	(60–74)	67	(58–74)	0.300
Obesity (BMI ≥ 30)	9	28.1%	16	24.6%	0.710
BMI (kg/m^2^)	27.8	(25.2–30.5)	27.2	(25.2–29.5)	0.583
Smoking (past or current)	16/25	64.0%	25/52	48.1%	0.190
Medical History					
Pulmonary ^#^	26	59.1%	36	55.4%	0.701
COPD	11	25.0%	7	10.8%	0.050
Diabetes	5	11.4%	15	23.1%	0.121
Chronic steroid treatment	17	44.7%	17	26.2%	0.053
Immuno-compromised *	2	4.4%	3	4.6%	0.966
CLINICAL COURSE					
Symptom onset to clinical admission (days)	7	(4–10)			
Symptom onset to ICU admission (days)	9	(7–13)	11	(9–15)	0.022 ^
SOFA score on ICU day 1	16	(15–17)	16	(15–17)	0.700
SOFA max on ICU	19	(18–20)	17	(16–20)	0.122
APACHE II	23	(18–27)	24	(21–32)	0.284
Lowest PaO_2_/FiO_2_ (mmHg)	123	(89–156)	157	(120–200)	0.002 ^
Steroids in ICU	9	29.0%	10	15.4%	0.117
RRT	15	46.9%	10	15.9%	0.001 ^
OUTCOME					
Symptom onset to CAPA (days)	17	(12–24)	Not applicable		
ICU admission to CAPA (days)	7	(4–12)	Not applicable		
Mortality at ICU day 30	23	39.7%	11	16.9%	0.005 ^

Data are presented as medians (IQR) or *n* (%). ^ = Significance at an alpha of 0.05 was found. *p* values comparing the possible/probable CAPA vs. non-CAPA cases were calculated using Mann-Whitney U (continuous variables) and Fisher or Chi-square tests when appropriate (categorical variables). Abbreviations: APACHE-II = Acute Physiology and Chronic Health Evaluation II; BMI, body mass index; CAPA = COVID-19-associated pulmonary aspergillosis; COPD, chronic obstructive pulmonary disease; COVID-19, coronavirus disease 2019; CRP, C-reactive protein; IQR, interquartile range; ICU intensive care unit; RRT, renal replacement therapy; SOFA, sequential organ failure assessment. ^#^ Any pulmonary disease prior to admission. * Immunocompromised as defined by the EORTC/MSG-ERC host factor criteria for invasive fungal disease.

**Table 2 jof-08-00096-t002:** Patient characteristics, clinical course, and outcome, by aspergillosis status and by study centre.

	Group A-CAPA Main Study Centre		Group B-CAPA Regional Study Centres		Group C Non-CAPA Main Study Centre		*p* Value
	(*n* = 29)		(*n* = 29)		(*n* = 65)		
Male	22	75.9%	18	62.1%	48	73.8%	0.836
Age (years)	70	(58–74)	69	(65–73)	67	(58–74)	0.525
Obesity (BMI ≥ 30)	9	31.0%	-		16	24.6%	0.515
BMI (kg/m^2^)	27.8	(25.7–30.9)	25.7	(24.2–27.2)	27.2	(25.2–29.5)	0.390
Smoking	13/20	65.0%			25/52	48.1%	0.198
Medical History							
Pulmonary ^#^	22	75.9%	-		36	55.4%	0.059
COPD	9	31.0%	-		7	10.8%	0.016 ^
Diabetes	2	6.9%	-		15	23.1%	0.060
Chronic steroid treatment	16	55.2%	-		17	26.2%	0.006 ^
Immuno-compromised *	1	3.4%	1/16	6.3%	3	4.6%	0.796
CLINICAL COURSE							
Symptom onset to clinical admission (days)	7	(4–14)	7	(6–9)			
Symptom onset to ICU admission (days)	9	(7–14)	9	(7–10)	11	(9–15)	0.198
SOFA score on ICU day 1	16	(15–17)			16	(15–17)	0.345
SOFA max on ICU					17	(16–20)	0.032 ^
APACHE II	23	(20–27)	18	(12–23)	24	(21–32)	0.516
Lowest PaO_2_/FiO_2_ (mmHg)	127	(93–161)	64	(52–76)	157	(120–200)	0.007 ^
Steroids in ICU	8	27.6%			10	15.4%	0.165
RRT	13	44.8%	-		10/63	15.9%	0.003 ^
OUTCOME							
Symptom onset to CAPA (days)	20	(14–26)	15	(12–20)	Not applicable		
ICU admission to CAPA (days)	8	(4–15)	5	(2–8)	Not applicable		0.110
Mortality at ICU day 30	15	51.7%	8	27.6%	11	16.9%	<0.001 ^

Data are presented as medians (IQR) or *n* (%). ^ = Significance at an alpha of 0.05 was found. *p* values comparing the CAPA (A) vs. non-CAPA (C) groups were calculated using Mann–Whitney U (continuous variables) and Fisher or Chi-square tests when appropriate (categorical variables). Abbreviations: APACHE-II = Acute Physiology and Chronic Health Evaluation II; BMI, body mass index; CAPA = COVID-19-associated pulmonary aspergillosis; COPD, chronic obstructive pulmonary disease; COVID-19, coronavirus disease 2019; CRP, C-reactive protein; IQR, interquartile range; ICU, intensive care unit; RRT, renal replacement therapy; SOFA, sequential organ failure assessment. ^#^ Any pulmonary disease prior to admission. * Immunocompromised as defined by the EORTC/MSG-ERC host factor criteria for invasive fungal disease.

**Table 3 jof-08-00096-t003:** CAPA classification and cultured Aspergillus associated with death.

		Deceased (*n*)	%	Survived (*n*)	%	*p*-Value
**CAPA CLASS**	Non-CAPA	11	16.9%	54	83.1%	0.016 ^
	Possible CAPA	9	36.0%	16	64.0%	
	Probable CAPA	14	42.4%	19	57.6%	
**ASP CULTURED ***	no	22	22.9%	74	77.1%	0.027 ^
	yes	12	44.4%	15	55.6%	

^ = Significance at an alpha of 0.05 was found. *p* values comparing the patients without CAPA (Non-CAPA) vs. Possible CAPA vs. Probable CAPA groups, and comparing Patients in which Aspergillus was cultured vs. not, were calculated Chi-square tests (categorical variables). * Aspergillus cultured in respiratory samples.

## Data Availability

Data is available in the clinical data centre of our hospital.

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
