# Peer review of "Regional Impact of COVID-19-Associated Pulmonary Aspergillosis (CAPA) during the First Wave"

_jof, 2022, doi:10.3390/jof8020096_

Round 1

Reviewer 1 Report

The manuscript by Bentvelsen et al describes an observational longitudinal study of CAPA incidence in critically ill COVID-19 patients in 2 centres in the Netherlands, with an analysis of factors associated with CAPA and the impact of CAPA on mortality/outcome. As such it is a useful addition to the current literature, and the increased mortality rates reported correlate well with recent reports from other European countries. I have a number of fairly minor points for the authors’ consideration, that would increase the utility of the current manuscript for other centres trying to diagnose and manage CAPA in COVID ICU patients.

Comments:

  1. Line 162 and throughout: spelling of anidulafungin and caspofungin
  2. Were the reported mortality rates adjusted for age/sex?
  3. What was the min, max and median time between ICU admission and isolation of fumigatus or formal diagnosis of CAPA? This is potentially important when interpreting the Kaplan-Meier plots. In both centres, CAPA vs control curves only start to separate around d5 after ICU admission. How does this correspond with CAPA diagnosis timing? Is poorer outcome determined by fungal co- superinfection, or are these infections occurring in patients who are already “more critically ill”? From the data in Table 1, one could infer that the CAPA diagnoses are being made around d8 post ICU admission? Is this correct? Similarly, significantly more CAPA patients required RRT, implying that they were generally more unwell. Please discuss these points.
  4. Also from table 1, two “entries” have an *, only one of which is defined in the legend.
  5. Was beta glucan testing used to aid in CAPA diagnosis? If so, what BDG positivity rates were seen in possible/probable CAPA patients as compared to in the control cohorts?

Author Response

The manuscript by Bentvelsen et al describes an observational longitudinal study of CAPA incidence in critically ill COVID-19 patients in 2 centres in the Netherlands, with an analysis of factors associated with CAPA and the impact of CAPA on mortality/outcome. As such it is a useful addition to the current literature, and the increased mortality rates reported correlate well with recent reports from other European countries. I have a number of fairly minor points for the authors’ consideration, that would increase the utility of the current manuscript for other centres trying to diagnose and manage CAPA in COVID ICU patients.

Comments:

1Line 162 and throughout: spelling of anidulafungin and caspofungin

2Were the reported mortality rates adjusted for age/sex?

3What was the min, max and median time between ICU admission and isolation of fumigatus or formal diagnosis of CAPA? This is potentially important when interpreting the Kaplan-Meier plots. In both centres, CAPA vs control curves only start to separate around d5 after ICU admission. How does this correspond with CAPA diagnosis timing? Is poorer outcome determined by fungal co- superinfection, or are these infections occurring in patients who are already “more critically ill”? From the data in Table 1, one could infer that the CAPA diagnoses are being made around d8 post ICU admission? Is this correct? Similarly, significantly more CAPA patients required RRT, implying that they were generally more unwell. Please discuss these points.

4Also from table 1, two “entries” have an *, only one of which is defined in the legend.

5Was beta glucan testing used to aid in CAPA diagnosis? If so, what BDG positivity rates were seen in possible/probable CAPA patients as compared to in the control cohorts?

Reply to Reviewer1.

We thank the reviewer for their time and effort and have taken the points in consideration as stated in the reply below.

1. Adjusted anidulafungin and caspofungin.

2. Mortality rates were not adjusted for age/sex.

3. Table 1 and 2 added: ICU to CAPA (days); Also, we agree on the RRT and worse P/F in CAPA could be confounding for worse outcome, independently from CAPA. We added this limitation to the discussion (line 198-201).

4. Table 1 * after Symptom onset to CAPA (days) was removed.

5. Beta-D-glucan testing was not used to aid in CAPA diagnosis, the test was not in use in the participating hospitals.

Reviewer 2 Report

Dear authors,

Thank you for this interesting paper. Please find my comments below.

Introduction

In the text, reference numbers should be placed in square brackets [ ], and placed before the punctuation.

Results

Line 162-163: Caspofungin and anidulafungin are written without “e” at the end (in English).

I think this information in the text about the cultured species, how the cases were diagnosed, which treatment was given would probably fit into another table (152-172). Was there a difference in mortality between the possible and probable cases?

Table 1 & 2

For me it’s not clear what “pulmonary medical history” means.

APACHE II is not explained in the abbreviations.

PaO2/FiO2 should is not mentioned in the text. Why is this not named Horowitz index? Why does it not have the unit mmHg? For me it is not clear how someone can have a Horwitz index of 1. Even in ARDS 1 seems impossible.

Example: PaO2/FiO2 = 60mmHg /1 = 60mmHg. To my knowledge the Horowitz index is NOT calculated PaO2/FiO2 = 60mmHg /100% = 0.6mmHg.

=> FiO2 of 100% means 1.0, => FiO2 of 50% = 0.5

Figures 1 & 2

Personally, I would not name the y-axis “cum” survival. It sounds strange. ;)

I don’t understand why 4 different curves are defined and the graph contains only 2.

Discussion

Did you have no cases of tracheobronchitis, or why did you have no proven cases?

You suggest a low threshold for the start of antifungal therapy, but why did not all possible/probable cases receive antifungal therapy? could this have had influence on the mortality?

Author Response

Introduction

1. In the text, reference numbers should be placed in square brackets [ ], and placed before the punctuation.

Results

2. Line 162-163: Caspofungin and anidulafungin are written without “e” at the end (in English).

3. I think this information in the text about the cultured species, how the cases were diagnosed, which treatment was given would probably fit into another table (152-172). Was there a difference in mortality between the possible and probable cases?

Table 1 & 2

4. For me it’s not clear what “pulmonary medical history” means.

5. APACHE II is not explained in the abbreviations.

6. PaO2/FiO2 should is not mentioned in the text. Why is this not named Horowitz index? Why does it not have the unit mmHg? For me it is not clear how someone can have a Horwitz index of 1. Even in ARDS 1 seems impossible.

Example: PaO2/FiO2 = 60mmHg /1 = 60mmHg. To my knowledge the Horowitz index is NOT calculated PaO2/FiO2 = 60mmHg /100% = 0.6mmHg.

=> FiO2 of 100% means 1.0, => FiO2 of 50% = 0.5

Figures 1 & 2

7. Personally, I would not name the y-axis “cum” survival. It sounds strange. ;)

 8. I don’t understand why 4 different curves are defined and the graph contains only 2.

Discussion

9. Did you have no cases of tracheobronchitis, or why did you have no proven cases?

10. You suggest a low threshold for the start of antifungal therapy, but why did not all possible/probable cases receive antifungal therapy? could this have had influence on the mortality?

Reply to Reviewer2.

We thank the reviewer for their time and effort and have taken the points in consideration as stated in the reply below.

1. Brackets for reference numbers were placed before punctuation.

2. Adjusted anidulafungin and caspofungin.

3. We agree that a table to clarify the differences in clinical outcomes of cases of probable CAPA vs. possible CAPA and with cultured Aspergillus fumigatus versus no cultured Aspergillus would give additional insight, we added table 3. Non-fumigatus Aspergillus was not found. Nearly all patients were treated in the main study centre.

4. Pulmonary history is assessed by the pulmonary physicians and comprises all pulmonary  conditions prior to admission, including COPD, Asthma, Sarcoidosis. We included a footnote in the tables.

5. We added APACHE II and CAPA to the abbreviations.

6. PaO2/FiO2 is the ratio this was erroneously a factor 100 to small and recalculated. We also revised the APACHE II scores and tested groups for significance (p-values).

7. Cumulative is adjusted and now written in full in the figures.

8. Survival curves per group and the symbol for censored cases are displayed. These are not ment as 4 curves, but 2 curves and 2 symbols

9. Tracheobronchitis was not proven as histological examination was inconclusive in two cases, ergo without proof of invasive fungal growth. In most cases no biopsy for histological examination was obtained.

10. The management of CAPA was new at the start of this trial. Untreated cases were diagnosed at a late stage or even after death. Current views are that CAPA is a recognised complication with increased mortality according to guidelines [8] and major studies [11].

Round 2

Reviewer 2 Report

Dear authors,

Thank you very much for your changes on the manuscript. It has clearly been improved.

Best regards